

# Observations and modelling of tidally generated high-frequency velocity fluctuations downstream of a channel constriction

Håvard Espenes[1,2], Pål Erik Isachsen[1,3], and Ole Anders Nøst[4,5]

[1]University of Oslo, 0315 Oslo, Norway
[2]Akvaplan-niva, 9007 Tromsø, Norway
[3]Norwegian Meteorological Institute, 0371 Oslo, Norway
[4]Nord University, 8026 Bodø, Norway
[5]Oceanbox.io, 9016 Tromsø, Norway

**Correspondence:** Håvard Espenes (hes@akvaplan.niva.no)

**Abstract.** We investigate data from an ADCP deployed in a constricted ocean channel showing a tidally dominated flow with intermittent velocity extrema during outflow from the constriction but not during inflow. A 2D numerical ocean model forced by tides is used to examine the spatial flow structure and underlying dynamical processes. We find that flow separation eddies generated near the tightest constriction point form a dipole pair which propagates downstream and drives the observed
intermittent flow variability. The eddies, which are generated by an along-channel adverse pressure gradient, spin up for some time near the constriction until they develop local low pressures in their centres that are strong enough to modify the background along-channel pressure gradient significantly. When the dipole has propagated some distance away from the constriction, the conditions for flow separation are recovered, and new eddies are formed.

## 1 Introduction

The flow through ocean channels, or straits, is often fast and associated with nonlinear dynamics – especially when forced by the barotropic tide (Stigebrandt, 1980; Farmer and Freeland, 1983). In such channels, flow separation eddies can emerge along the channel walls (Kundu et al., 2015). As shown by Wells and van Heijst (2003), tidally-generated flow separation eddies can pair up to form dipoles near the channel exit. These can then self-advect away from the channel, even on the reverse tide. Model studies have shown how these vortex structures have the potential to trap fluid and thus drive significant net tracer transport

through a channel (Feng et al., 2019; Nøst and Børve, 2021; Børve et al., 2021). So a good understanding of the nonlinear processes that govern the life cycle and behaviour of such vortices is required if one wishes to make accurate estimates of tracer dispersion in a complex coastal zone.

Flow separation dipoles in the field have been well documented through various drifter and shipboard ADCP-based studies (Fujiwara et al., 1994; Old and Vennell, 2001; Whilden et al., 2014). However, analyses of dipole dynamics from fixed obser-

vations has received less focus. Easton et al. (2012) presents vertically-averaged flow observations recorded by an instrument moored downstream from a channel in the Pentland Firth, Scotland. This data set shows high-frequency flow oscillations on the outflow phase of the tide but, importantly, not on the inflow phase. The authors hypothesize that an unidentified problem



with their mooring setup could be the cause of the oscillations, essentially neglecting processes that do not appear to be tidal waves. However, the asymmetry in their mooring response suggests that what they picked up might instead be actual variability
caused by flow separation eddies.

The theoretical basis for such eddies originates in a study by Stommel and Farmer (1952), who first approached a dynamical description of oceanic channel flows by noting the distinct asymmetry between a channel's inflow and outflow side. Currents flowing into the channel resemble a potential flow, whereas currents flowing out separate from the channel walls to form a jet. Bernoulli's principle, along with the local effects of side-wall friction, can be used to explain this phenomenon. To conserve the
barotropic volume flux through a channel, i.e. to keep $V_f = A \cdot u$ constant, the flow speed ($u$) must either increase or decrease as the channels' cross-sectional area ($A$) changes. To do so, some of the fluid's potential energy is converted to kinetic energy as water flows into the channel and speeds up. Conversely, kinetic energy is converted to potential energy as water flows out of the channel and slows down. This results in a dynamic low pressure, i.e. a dip in the sea surface, at the tightest point of the channel and a resultant pressure gradient force directed toward that point from both sides. On the downstream side, the pressure gradient
force is thus against the flow direction. When acting in conjunction with side-wall friction, this *adverse* pressure gradient force may cause some water parcels to stall near the channel walls and even flow back toward the constriction. This situation is known as flow separation, resulting in flow with locally closed streamlines—the flow separation eddies—and a jet shooting out between these (Kundu et al., 2015).

Studies exist that have examined the force balance of real channel flows in light of this theoretical framework. Vennell (2006)
used a ship-mounted ADCP to argue that the dynamics within a channel were quasi-stationary throughout an entire half tidal cycle (with the flow being in one direction), dominated by the advection of momentum balancing an along-channel pressure gradient force. Their results agree well with Hench et al. (2002) and Hench and Luettich (2003) who, based on model results, argue that the momentum balance is quasi-stationary on ebb/flood on both sides of both an idealised and a realistic channel. Interestingly, Hench et al. (2002) tuned the eddy viscosity in their model and found that weak viscosity exited transient eddies
in their simulations. However, they too dismissed these eddies as (model) noise.

The present study offers an alternative interpretation of the noisy signals reported by both Hench et al. (2002) and Easton et al. (2012) by analyzing ADCP data collected from another tidally dominated strait, this one located in Northern Norway. The observations contain a similar high-frequency velocity signal which we can not attribute to instrument errors. The observations are accompanied by high-resolution numerical modelling. We formulate two objectives: 1) to use realistic model simulations
to identify flow patterns capable of generating similar current time series to those captured by the ADCP and 2) to use realistic and idealized simulations to investigate the underlying dynamics. The results indicate that the observed high-frequency flow variability is indeed tied to the generation of flow separation eddies and that the earlier claims of a quasi-stationary force balance must be nuanced.





## 2 Methods

### 2.1 Observations


We analyse the flow in the Tromsøysund strait in Northern Norway which connects the 60 km long fjord Balsfjord to the open ocean (see Figure 1). The observations were collected with an acoustic doppler current profiler (ADCP) deployed roughly one kilometre north of the narrowest point of the constriction. The instrument was an Aanderaa Seaguard II, configured to record data for 2.5 minutes and store averages of these recordings in intervals of ten minutes, aiming to reduce the influence of

instrument-induced noise and reduce battery consumption. The instrument was deployed in a rigid bottom frame at the end of the winter season, from March to April 2017.

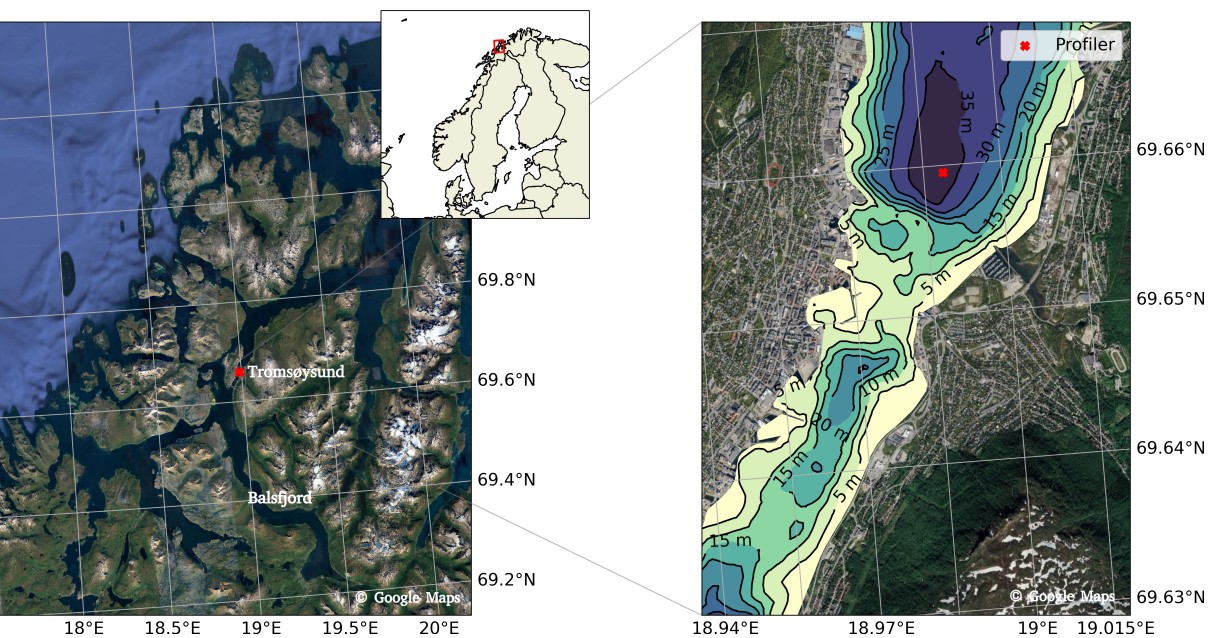

**Figure 1.** The study domain with the location where we recorded the flow in Tromsøysund marked with a red cross. The right panel zooms in on Tromsøysund with bathymetry shaded and contoured. Background land information has been collected from Cartopy Google map tiles (Google, 2021).

Below we will be presenting and discussing the 'along-channel' and 'across-channel' components of currents, where the along-channel direction is shown with a line in Fig. 2. These two flow components were found by rotating the original east-west ($u$) and north-south ($v$) flow components by angle $\theta$ between east-west and the along-channel direction as

$$V_{along} = u \cdot cos(\theta) + v \cdot sin(\theta)$$
$$V_{across} = -u \cdot sin(\theta) + v \cdot cos(\theta)$$

(1)



## 2.2 Numerical modelling

We use the Finite Volume Community Ocean Model (FVCOM, Chen et al., 2003). This is a prognostic, free-surface three-dimensional primitive equation ocean model which solves the integral form of the equations on an unstructured triangular horizontal grid. In this study, we use a 2D configuration of FVCOM that solves the depth-averaged momentum and continuity equations for a constant-density fluid. FVCOM calculates momentum advection using a second-order accuracy flux scheme (Chen et al., 2006; Kobayashi et al., 1999). We used a Smagorinsky (1963) closure scheme for the horizontal diffusion of momentum to suppress velocity fluctuations of length scales comparable to the mesh resolution. The Smagorinsky coefficient ($C_{smag}$) was tuned by comparing the modelled flow with the observed flow at the ADCP (see below), and the best fit was found when this coefficient was set to 0.2. Furthermore, we used a quadratic bottom friction parametrization $(\tau_{b,x}, \tau_{b,y}) = \rho_0 C_d |\mathbf{u}| \mathbf{u}$ where $C_d = \frac{9.81}{2500} D^{-1/3}$. The governing equations are integrated in time using a modified explicit fourth-order Runge-Kutta time-stepping scheme (see Chen et al., 2003, for more details). This choice of model physics and parameter settings was used for two tidally-forced simulations, one with realistic and one with idealized geometry.

### 2.2.1 Realistic simulation

We used the mesh-generator distmesh (Persson and Strang, 2004) to create a mesh focused on Tromsøysund, consisting of 747.761 triangles (cells) and 393.422 triangle corners (nodes). The resolution varied from 13 m in the Tromsøysund to 8 km at the open boundary (see Fig. 2). The domain was bounded by land defined by coastline polygons obtained from The Norwegian Mapping Authority (Kartverket, 2022). The smallest triangle angle in the domain was 30 degrees and the largest was 117 degrees. No triangles had more than one side facing the model boundary. The bathymetry was also retrieved from the Norwegian Mapping Authority high-resolution bottom bathymetry database.

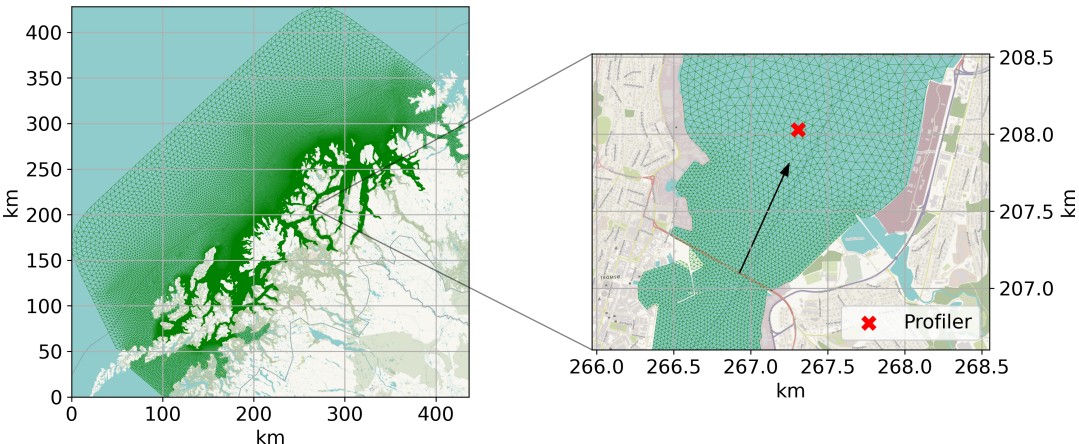

**Figure 2.** Model mesh. Zoom-in on Tromsøysund to the right. The black arrow to the right shows the along-channel direction. Georeference from © OpenStreetMap contributors 2023. Distributed under the Open Data Commons Open Database License (ODbL) v1.0



The model was forced with sea surface height (SSH) data of 13 tidal harmonics (M2, S2, N2, K2, K1, O1, P1, Q1, Mf, Mm, M4, MS4, MN4) extracted from TPXO7.2 (Egbert and Erofeeva, 2002) and interpolated to the open boundaries. We used TPXO7.2 since the model performed better in Tromsøysund using this dataset than when using the newer TPXO9-atlas-v5. FVCOM then computes velocities at the open boundary using a reduced set of equations (Chen et al., 2003). We ramped the model up from rest and a flat sea surface to full forcing over 30 days.

To assess the model performance, we compared the model results to sea surface elevation measurements from 18 mooring stations in the Tromsø region. The moorings were made up of a mixture of point measurement devices (Aanderaa RCM) and ADCPs (Aanderaa Seaguard II and Nortek Aquadopp); all were fitted with pressure sensors that operated for 1–2 months to cover two or more spring-neap cycles. Figure 3 shows the mooring locations and a scatter plot of the observed and modelled SSH amplitude of M2, S2, N2 and K1 tidal constituents after these were calculated with the UTide harmonic analysis software

(Codiga, 2011). The semi-diurnal constituent (M2, S2, N2) amplitudes increase going northward and are an order of magnitude bigger than the diurnal constituent (K1); this is consistent with findings by Gjevik et al. (1994). The modelled M2 amplitude is slightly underestimated, and the observed spread of the other components is larger than the modelled spread. But we take the relatively low scatter around the one-to-one line as an indication that the model is reproducing the tides in the region sufficiently well for our process-oriented study.





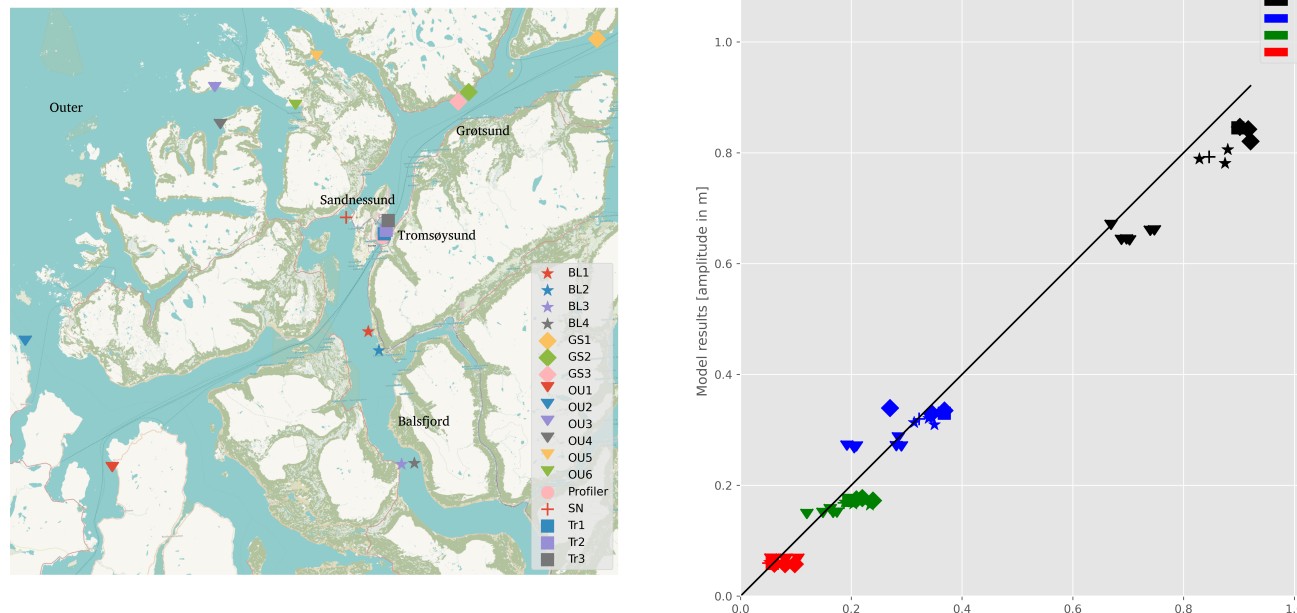

**Figure 3.** Comparison between tide SSH amplitude of four leading harmonics observations. The left hand panel shows the position of the moored instruments and depth of deployment. The mooring stations are labelled by the geographical subdomains they were deployed in; "BL" (from the Balsfjord, south of Tromsø), "GS" (from Grøtsundet, north of Tromsø), "Tr" (from the Tromsøysund) and "SN" (from the Sandnessund). Stations outside of the fjord system are labelled "OU". The right hand panel compares the modelled tidal elevation to the observations. Georeference from © OpenStreetMap contributors 2023. Distributed under the Open Data Commons Open Database License (ODbL) v1.0

### 2.2.2 Idealized simulation

To help isolate the key dynamics in a more idealized setting, we used the same FVCOM channel setup as in Nøst and Børve (2021). The domain is shown in Fig. 4 and consists of a 1 km wide channel which cuts through a peninsula. The channel has a maximum depth of 100 m deep in its centre and is 5 m deep near the side walls. The model is forced at the open boundary with a semi-diurnal tidal wave with SSH amplitude of 10 cm, and velocities at the boundary are again calculated by the reduced physics algorithm as used in the realistic simulation.





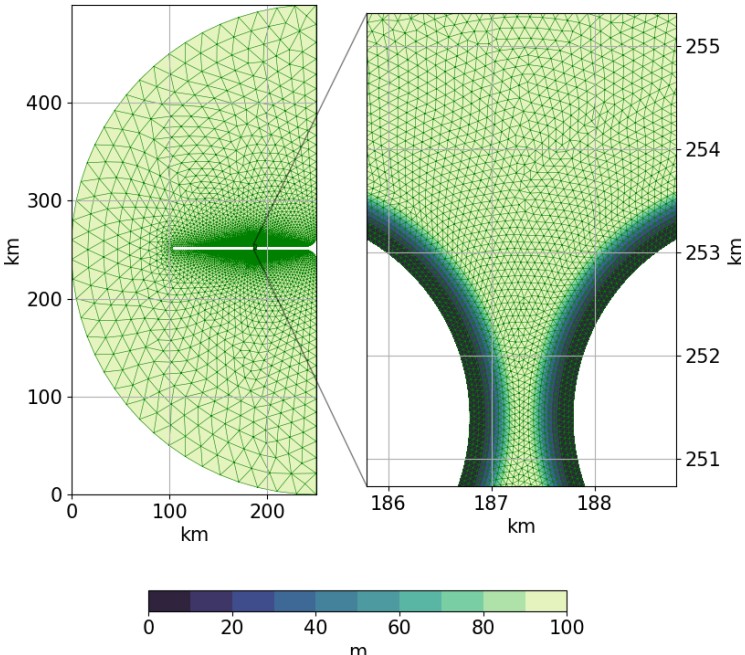

**Figure 4.** Idealized model mesh with shaded bathymetry as background

## 3   Results

### 3.1   Observed and modelled flow in Tromsøysund

The currents in Tromsøysund, measured by the ADCP at all depth levels over approximately three diurnal tidal cycles during spring tide, are shown in Fig. 5. The velocity components have been rotated, as described above, so what we see here is the flow approximately in the along-channel direction. Positive values indicate flow out of the constriction (predominantly northward), whereas negative values are into the constriction (southward). A semi-diurnal signal dominates the time series, but the flow is asymmetric, being stronger on outflow than inflow. In addition, the record reveals high-frequency variability, which appears to be more predominant on outflow than inflow. Finally, both the tidal signal, as well as the high-frequency variability are predominantly barotropic, lending strong support to the idea of treating the flow as two-dimensional in the model simulations.





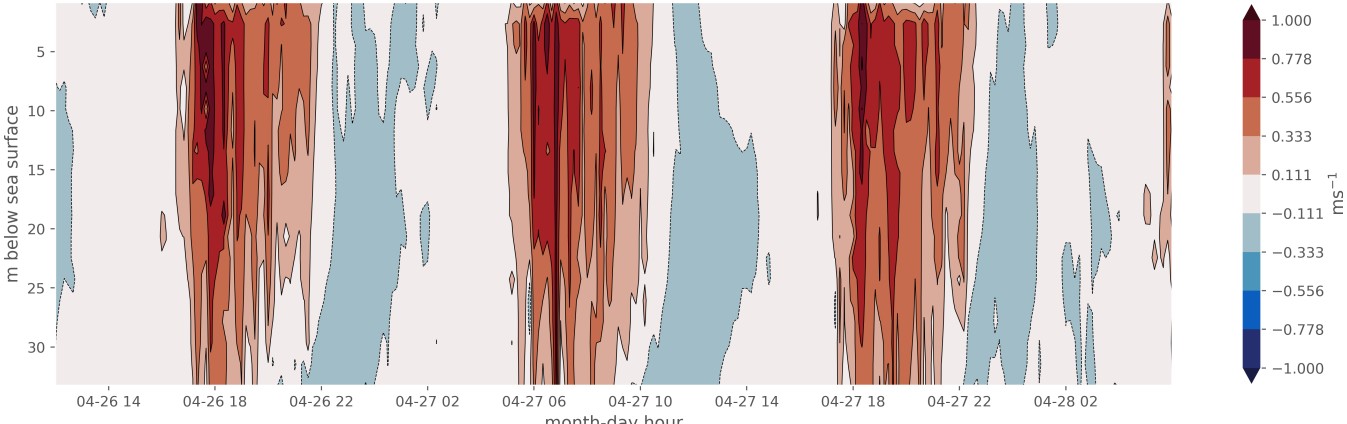

**Figure 5.** The along-channel flow component as a function of time and depth, as measured by the ADCP instrument in Tromsøysund.

Figure 6 shows the observed velocity both in the along-channel (left-panel) and across-channel (right-panel) directions, now after depth-averaging. So positive values here indicate flow with predominantly northerly and easterly components, respectively. Each reading is indicated with a red dot. In addition, a blue line indicates the pure tidal components of the flow, reconstructed using the UTide tidal harmonic analysis package (Codiga, 2011). We now see even more clearly how the along-channel flow (left panel) is asymmetric, stronger on outflow than on inflow. The harmonic analysis picks up the asymmetry at tidal frequencies and reproduces the southward flow (inflow). But on northward flow—from the constriction and toward the ADCP—the observations show high-frequency velocity spikes of about 20 cm/s superimposed on the tidal signal. The fact that these are systematically observed on only northward tidal flow suggests that they are associated with tides rather than, e.g. atmospheric fluctuations. But being very high-frequency, the harmonic analysis does not capture them.

The across-channel (right panel) flow pattern is similar to the along-channel flow in that the harmonic analysis captures the dynamics extremely well on the southward but not on the northward tide. Notably, whereas the tidal signal is strongest in the along-channel direction, the high-frequency variability is fairly isotropic. Thus, we will subsequently focus on the high-frequency velocity fluctuations in the along-channel direction when comparing them with the model results below.

Figure 7 compares the model and observations for the same three semi-diurnal tidal cycles. The depth-averaged along-channel flow component is shown in the left panel, while SSH is shown in the right panel. The numerical model captures the along-channel flow asymmetry well, including a clear predominance of high-frequency spikes on the out-flowing phase of the tide. In this simulation, the amplitude of the spikes is generally somewhat higher than in the observations, but this can partially be tuned by choice of model bottom friction and viscosity parameters (see Discussion). The modelled sea surface elevation signal also tracks the observations fairly well, as suggested by the more broad comparison shown in Fig. 3, even if small amplitude and phase offsets can be detected.



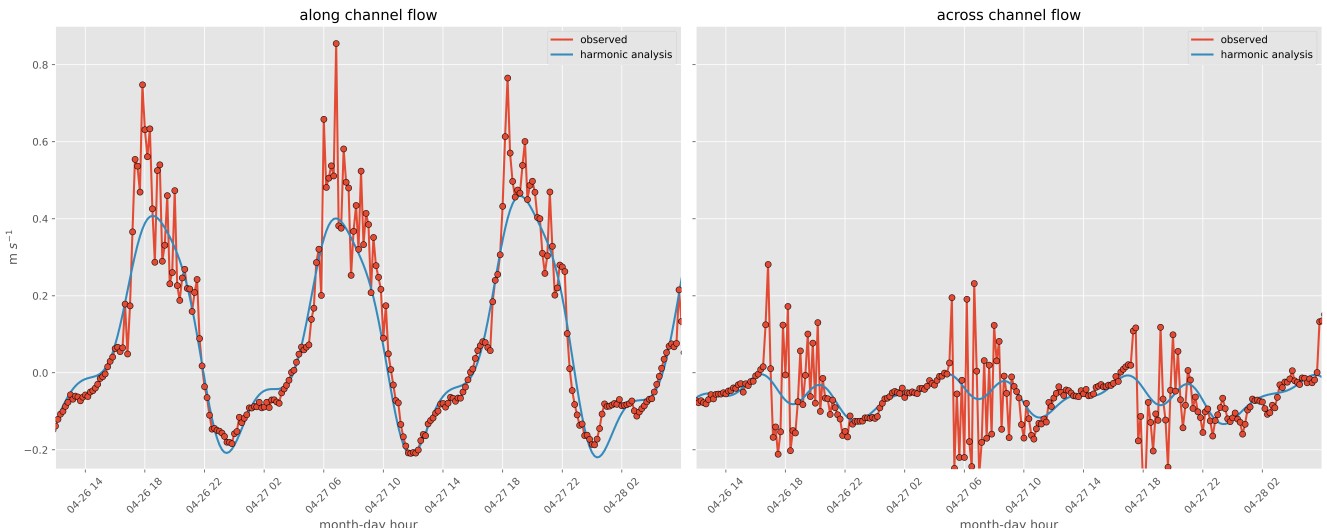

**Figure 6.** Depth-averaged velocity components in along-channel (approximately northerly) and across-channel (easterly) directions, as measured by the ADCP. Dots indicate each actual observation. Blue lines are reproductions of the tidal contribution based on harmonic analysis.

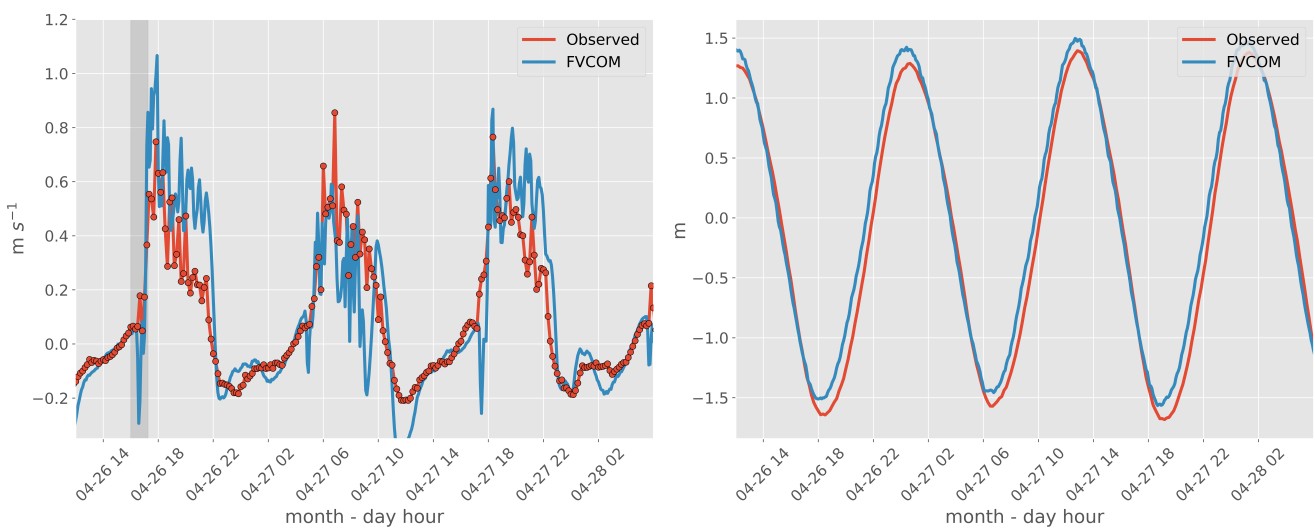

**Figure 7.** Observed and modelled along-channel velocity (left) and sea surface elevation (right) 26 April at 12:00 2017 to 29 April at 12:00 2017. Grey shade indicates the time period studies in detail in Figure 8

Encouraged by the model-observation comparisons, we look at model fields in more detail to investigate the source of the high-frequency fluctuations. Figure 8 shows the model flow field in Tromsøysund for nine-time snapshots on 26 April 2017





(the period is indicated by dark grey in Fig. 7). The bottom panel of the figure shows a time series of the along-channel flow extracted from the nearest mesh triangle to the ADCP (indicated with a white star in the above nine snapshot panels).

**Figure 8.** Snapshots of the flow in Tromsøysund, showing streamlines and flow speed (colour). Panel A is from 16:00 on the 26th of April. Relative to A, B is +15 mins, C is +30 mins, D is +40 mins, E is +50 mins, F is +1 hr, G is +1 hr 5 mins, H is +1 hr 10 mins, and I is + 1 hr 15 mins. The lower panel shows a time series of the along-channel flow at the grid location closest to the ADCP (white star). Dots and bars of different colours indicate when the snapshots in the top nine panels were taken. Georeferences from © OpenStreetMap contributors 2023. Distributed under the Open Data Commons Open Database License (ODbL) v1.0





Panel A, taken only a few minutes after the tide has started to flow northward, shows that two flow-separation eddies have
already formed just downstream of the constriction. Panel B shows the situation 15 minutes later (+ 15 mins) when the eddies
have intensified. They have now detached from their generation site and started to move downstream. The combined velocity
field of the dipole pair concentrates and guides the jet in between them. Another 15 minutes later, in panel C (+30 mins), the
dipole has moved further downstream, and the left-hand eddy, which rotates counter-clockwise, has begun to influence the
strength and direction of currents where the ADCP is located. As this left-hand eddy passes just to the right of the ADCP, it
reverses the currents sampled at this location, as also seen in panel D (at +40 mins and as outlined in the lower time-series
panel). During this period and, in fact, throughout the remaining panels, the right-hand eddy is fairly stationary, appearing to
be trapped by features in the coastline.

The initial dipole pair is particularly strong and noticeable. But a series of secondary eddies also form as soon as the initial
pair has left the generation site. In particular, a string of new counter-clockwise eddies form on the left-hand side, captured
in panels C, E and G. These, in turn, also propagate towards and past the ADCP, impacting the flow recorded by the fixed
instrument there. At least one new clockwise eddy can also be seen spinning up on the right side of the channel in panels G to
I.

So a main take-home message from Fig. 8 is that the high-frequency variability recorded by the ADCP instrument in Trom-
søysund reflects the chaotic flow related to the flow-separation eddies and jet. The jet is concentrated between eddy dipole
pairs, and its strength can be comparable to the flow in the narrowest part of the constriction itself. Multiple secondary eddies
can form after the initial dipole detaches, and the relative positioning of the eddies, both primary and secondary, decides how
the jet meanders. Details of the channel geometry have a large impact, and in Tromsøysund, eddies forming along the right
side are partially trapped by the coastline shape. In contrast, eddies to the left easily propagate down the channel.

A second point to note is that the observed progression of events from primary eddy formation via detachment and dipole
advection to the generation of secondary eddies suggests that the dynamical balances in the constriction may not be as stationary
through the outflow period as suggested by some earlier studies. To examine the underlying dynamics in more detail, we,
therefore, proceed to take a look at the along-channel pressure field. And to help isolate the relevant dynamics, we first turn to
the simulations of flow in the much more idealized channel geometry.

## 3.2 Dynamical evolution

Figure 9 shows snapshots of the flow field from an out-flowing tide in the idealized run. The flow is initially similar to a
Bernoulli flow (panel A). The streamlines converge on the inflow side, the maximum speed is in the centre of the channel, and
streamlines diverge on the outflow side. A large counter-clockwise eddy downstream of the constriction is a remnant from an
earlier tidal cycle. Panel B, taken one hour later (+ 1 hr), then shows the flow state shortly after flow separation, and we see
that eddies have formed on both sides of the channel, similar to the initial dipole in Tromsøysund. This dipole pair modifies
the flow by concentrating it in a narrow band—the jet—exiting the channel. In panel C (+2 hrs), the dipole has pinched off the
constriction and begun to move downstream. It continues to propagate away from the channel in subsequent time slices. Panel
D (+2 hrs 30 mins) then shows that a secondary eddy develops to the right in the channel after the dipole has left. New eddies




can also form in panels F and G (at +3 hrs 30 mins and 4 hrs). So the development of both primary and secondary eddies and dipoles are also captured in this simulation. And again, we see that the jet strength, especially when guided between a dipole

pair, is comparable and can even surpass the flow strength in the narrowest part of the constriction itself.

**Figure 9.** Flow field evolution in the idealized channel run. Relative to panel A, B is +1 hr, C is +2 hrs, D is +2 hrs 30 mins, E is +3 hrs, F is +3 hrs 30 mins, G is +4 hrs, and F is +4 hrs 30 mins.

As the dipole guides the jet, it is not entirely obvious how to separate the two. So one can also say that the flow speed associated with the dipoles themselves is comparable in strength to the speed at the constriction both in the idealized and realistic simulation. This suggests that the pressure perturbations associated with those eddies may also be comparable to the channel-scale dynamic pressure—and adverse pressure gradient—which causes flow separation in the first place. We now look

for evidence of this by examining how the pressure field (SSH) near the constriction changes in response to the presence of eddies.



Figure 10 shows the along-channel pressure gradient force with streamlines plotted on top. Red and blue colours here indicate a pressure force up and down the channel, respectively. The time interval is somewhat shorter than in Fig. 9 as we now wish to examine the dynamics around the initial dipole pair primarily. Panel A again shows the situation just after slack water, when a weak large-scale positive pressure gradient has started to accelerate the fluid through the channel. At this early time, the local dynamic pressure gradient is weaker than the large-scale gradient that actually drives the flow. Panel B shows the state 30 minutes later (+30 mins). An adverse pressure gradient has formed and acts to decelerate the flow on the outflow side of the channel, and there is already some sign of flow separation building up near the left channel wall. In panel C, at +1 hr, two distinct eddies have formed, one on either side of the channel. Being in cyclostrophic balance, their along-channel pressure gradient signals are adverse on the downstream flank but favourable (in the direction of the large-scale forcing) on the upstream flank. In panel D, at +1 hr 15 mins and after the eddies have grown in size and strength, the eddy pressure anomalies influence the larger-scale adverse pressure gradient around the channel exit considerably, reversing its sign near the walls. The two vortices still spin up at the location of their generation at this stage. But after another 15 minutes, in panel E (+1 hr 30 mins), the eddies detached from the sidewalls and started to approach each other. The favourable pressure gradient force tied to the dipole extends across the entire channel width. It is after this time that the dipole pair begins to translate and leave the constriction (panels F–H). Importantly, as the primary dipole pair have travelled slightly downstream, a distance amounting approximately to their diameter, the background adverse pressure gradient again dominates near the channel constriction. And in panel G (+2 hrs 30 mins), a secondary eddy has formed along the right-hand wall (corresponding to panel D in Fig. 9). This new eddy then commences travelling downstream in panel H (+3 hrs 15 min).





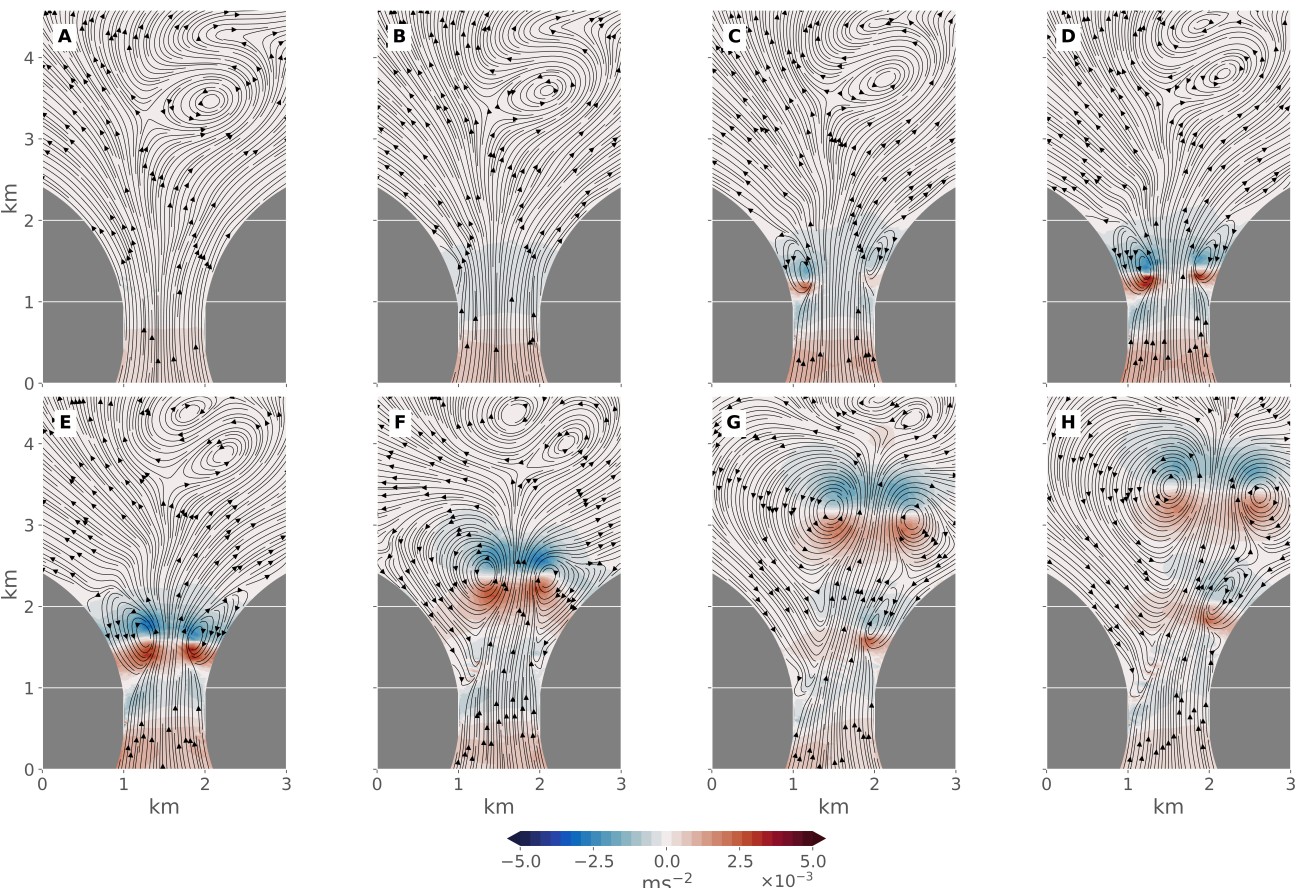

**Figure 10.** Along-channel pressure gradient during eddy growth. Relative to A, B is +30 mins, C +1 hr, D +1 hr 15 mins, E +1 hr 30 mins, F +2 hrs, G +2 hrs 30 mins and H is +3 hrs 15 min.

So Fig. 10 suggests that the initial vortex pair detaches from the generation region when its pressure anomalies can locally cancel the adverse pressure gradient across the channel width. To examine this relationship, we form a time series of the along-channel pressure gradient and along-channel velocity, now averaged over a transect which crosses the channel approximately where the vortices originally form. Figure 11 shows the result. A little more than two tidal cycles are shown, and grey shading indicates the period corresponding to Fig. 10. When interpreting the time series, it is worth noting that the pressure gradient force here (as well as in Fig. 10) is the *total* force which includes i) the large-scale background pressure force due to the imposed tidal wave, ii) the dynamic force which reflects the exchange between potential and kinetic energy along the channel and iii) the force anomalies due to the eddies.

The cross-channel averaged pressure gradient force is directed toward the constriction (negative) on southward flow, as expected as both the large-scale tidal and the channel-scale dynamic pressure gradients point in the same direction. The velocity signal is then smooth, acting as a potential flow, as seen explicitly in the above-mentioned fields. Around the subsequent slack water, there is a brief period when the large-scale forcing dominates and sets up a pressure gradient force out of the constriction



(positive). But the local dynamic pressure soon dominates and establishes an adverse pressure gradient against the large-scale flow through the channel. Around this time, the cross-channel averaged velocity decreases, reflecting reverse flow associated with the spin-up of the primary dipole pair. Note that the reduced cross-channel averaged velocity doesn't need to imply a reduced total volume flow since the reverse flow occurs near the channel walls where the depth is reduced. Then follows a sharp reversal in pressure gradient, a spike, which coincides with an increase in the averaged flow speed. The abrupt transition reflects the detachment and propagation of the initial vortex pair away from the generation region—and the transect. Subsequent noise in both pressure force and velocity signals presumably reflects secondary eddy formation, detachment and propagation.

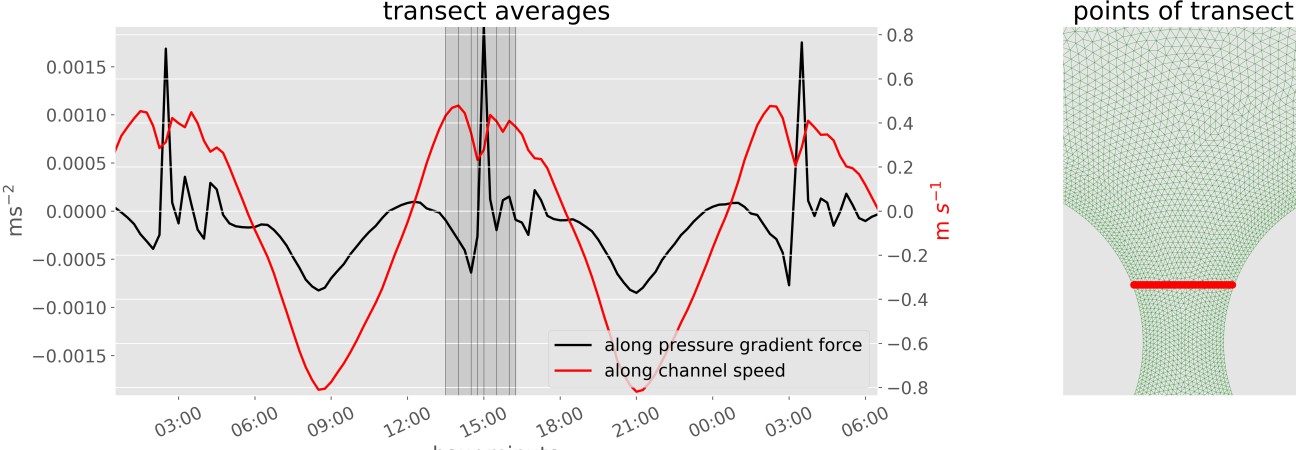

**Figure 11.** Left: Across-channel averaged along-channel pressure gradient (black line) and along-channel velocity (red line). The dark shaded area indicates the time-span of panels shown in Figure 10. The vertical, grey lines, indicates values for each panel in Figure 10. Right: Red dots illustrate that the transect was extracted from a location downstream from where the channel expands. The absolute time here is arbitrary.

The results thus indicate that the model never reaches a quasi-steady adverse pressure gradient force during outflow conditions due to the dynamic signals imposed by both primary and secondary flow-separation eddies. We now return to Tromsøy-sund to see if the same force pattern can be found there. Snapshots of the along-channel pressure gradient force and streamlines there from the realistic simulation are shown in Figure 12. The panels are taken from the same tidal cycle as in Figure 8 but now again focus on the period around the initial dipole pair. Panel A shows the situation just after slack tide. The along-channel pressure gradient downstream of the narrowest point of the channel is positive, reflecting the large-scale pressure gradient driving the tidal signal. The streamlines at this point in time suggest potential flow. A region of reversed pressure gradient southward of the narrowest constriction is the dynamic imprint of an older vortex situated there. 20 minutes later (panel B, +20 mins), the flow strength through the channel has increased, and the pressure gradient force downstream from the channel centre has turned adverse. But the streamlines still mostly follow the channel except near the first flow-separation eddy, which has formed along the left boundary. At +30 mins (panel C), the pressure gradient force is strongly adverse where the channel expands, and a flow-separation eddy has also formed on the right side. At +40 mins (panel D), the dynamic pressure signal from the two





eddies extends across the channel, creating a favourable gradient in their wake. At this point, the counter-clockwise vortex to the left starts to move downstream. The remaining panels show a complex pressure force pattern which is harder to interpret. But between +55 mins and +1 hr 5 mins (panels E and F), we observe a recovery of a strong adverse pressure gradient and the generation of a secondary eddy on the left-hand side. After this time, more secondary eddies form, at least on the left side. 235 And these then appear to be associated with the development of alternating bands of favourable and adverse pressure gradients along the jet path.

**Figure 12.** Along channel pressure gradient. Relative to Panel A, B is +20 mins, C is +30 mins, D is +40 mins, E is +55 mins, F is +1 hr 5 mins, G is +1 hr 15 mins, H is +1 hr 25 mins and I is +1 hr 35 mins. Panel E and F correspond to Panel B and C in Figure 8.





Finally, as for the idealized channel, we average fields across the strait, approximately where flow separation occurs. Figure 13 shows the resulting time series of along-channel pressure gradient force and velocity. These averaged fields are also noisier than the corresponding fields extracted from the idealized channel run, but a few key features seen earlier can be rec-

ognized. On southward inward, when the large-scale forcing and dynamic Bernoulli response have the same signs, the total pressure signal is fairly smooth, reflecting potential flow. After slack tide, at the very beginning of northward outflow, the net force tends toward negative values—towards the constriction—as the flow speed increases. But then comes a very strong positive, i.e. favorable, pressure force spike. This behaviour on outflow, the build-up of an adverse pressure gradient interrupted by a favourable pressure gradient spike, is seen in all the shown cycles. The spike is not, however, associated with the concurrent

reduction in flow speed that we saw in the idealized case. The difference here, we speculate, is tied to the volume conservation constraint and, by implication, details in the bottom topography profile across the channel. After the initial spike, the channel pressure force fluctuates between positive and negative values in quite a nosier manner, as suggested by the later snapshots of Figure 12. So, despite the more noisy situation here, it is clear that an adverse pressure gradient is not stably present throughout the outflow phase of the tide.

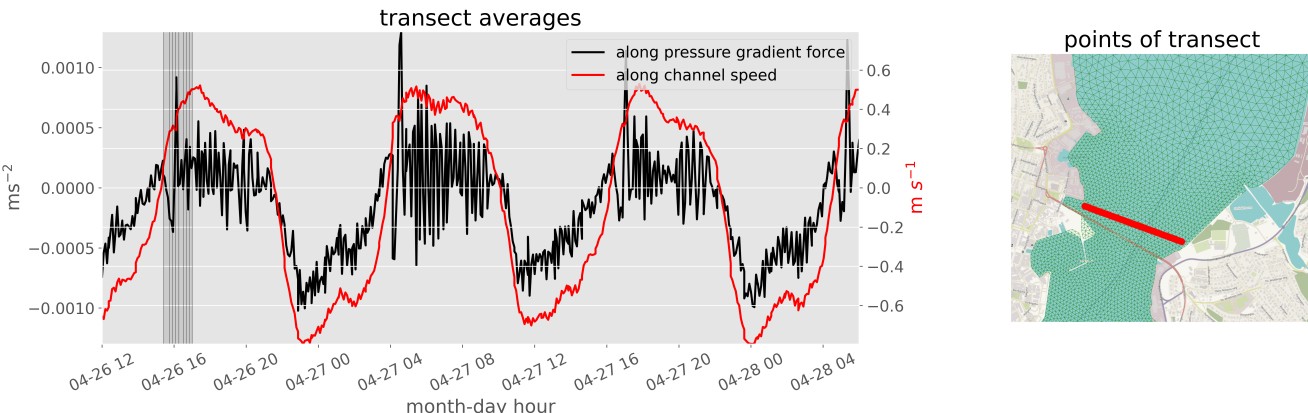

**Figure 13.** Left panel: Cross-channel averaged along channel pressure gradient force in Tromsøysund. The grey shading indicates the timespan of all panels in Figure 12, and each vertical line indicate each individual panel. Right panel: Positions used to compute the transect. Georeference from © OpenStreetMap contributors 2023. Distributed under the Open Data Commons Open Database License (ODbL) v1.0.

## 4 Discussion

### 4.1 Pressure gradient at the entrance

Figure 14 outlines what we believe are the fundamental features of the dynamical fields studied above. The left panel shows a fictitious sub-critical and in-viscid Bernoulli flow through a constricted channel. Here, the sea surface dips at the centre of the constriction due to the conversion from potential to kinetic energy. On top of this nonlinear response is a large-scale pressure




gradient which drives the flow in the first place (not shown). However, the smooth potential flow shown in the left panel is unattainable when accounting for side-wall friction. In reality, friction and the adverse pressure gradient force downstream of the constriction work against the flow, and the flow will separate from the side walls (Kundu et al., 2015). The separation generates circular motions—eddies—on both sides of the constriction, as shown in the right panel.

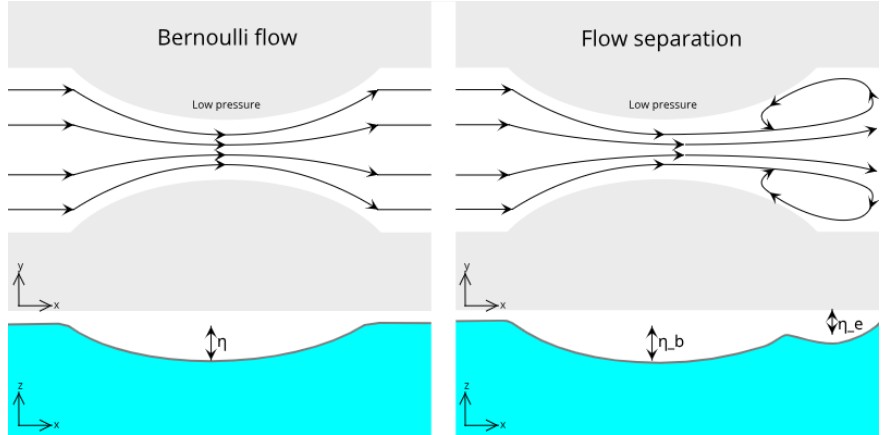

**Figure 14.** Sketch of streamlines and sea surface perturbation in a pure Bernoulli flow and with flow separation and eddies downstream

As the cyclostrophic eddies spin up, low-pressure anomalies develop in their centres. Our model results indicate that the
eddies pinch off the constriction when the eddy-induced pressure anomalies are strong enough to change the direction of the pressure gradient around the position where the flow is separated initially. Once the direction of the pressure gradient force is reversed locally, the conditions which caused flow separation in the first place have been violated. The eddies then detach from the walls and translate down the channel, either advected by the jet or self-advecting as a dipole (see below).

### 4.2 Implication for startup and dipole velocity

Our interpretation of the model results above may offer an alternative view into two closely related topics related to flow-separation dipoles, namely startup time and dipole velocity. The startup time, as defined by Afanasyev (2006), is the time between initial vortex formation and the moment when the dipole starts translating. Earlier scaling arguments have focused on the build-up of sufficient vorticity in the eddies, while here, we argue that the dipole starts translating when the eddy pressure field is strong enough to dominate across the strait in such a way that the necessary conditions for flow separation are removed.
Thus, the evolution of the eddy pressure field determines the startup time. Thus, we suggest that the vorticity and pressure descriptions should be complementary. Simply put, since the eddies are in cyclostrophic balance, their vorticity and pressure fields are tightly connected. Therefore, an eddy building up its vorticity, fed by the strong vorticity front created by flow separation, will also have a pressure field that builds up. The added value of the pressure analysis used here is the connection to the dynamic pressure gradient force, which controls flow separation in the first place.





Thus, as we have seen, when the dipole moves away from the strait, flow separation conditions are restored, leading to the formation of the next dipole pair. Hence, multiple dipole pairs can form over one half tidal cycle. The time between new dipole pairs or, alternatively, the frequency of dipole formation should be determined by the startup time plus the time it takes for a dipole to move a distance sufficient for the flow separation conditions to be restored. This distance is likely related to the vortex diameter, and the translation time scale then depends on the vortex size and the dipole translation velocity.

But are dipoles in a tidal strait advected passively by the jet after detaching, or do they self-advect? Afanasyev (2006) found that dipoles formed in the laboratory moved with speed equal to half the jet velocity $U_{max}$. And Nøst and Børve (2021), having simulated and studied flow in several ideal tidal straits of different geometry, found that flow-separation dipoles moved with a velocity $U_{dip} = (a/b)U_{max}$, where $a$ is the eddy core radius, and $b$ is the core separation distance. Assuming, for simplicity, a jet which increases linearly from zero at the channel walls to $U_{max}$ in the centre, the results of Afanasyev (2006) and Nøst

and Børve (2021) thus agree for vortices that each have a diameter which is half the channel width. However, Nøst and Børve (2021) also found that the same dipoles followed

$$U_{dip} = \frac{|\Gamma_1| + |\Gamma_2|}{2\pi b}, \tag{2}$$

where $|\Gamma_1|$ and $|\Gamma_2|$ are the magnitudes of the circulation in each of the two eddies. The expression, incidentally, is twice that which is traditionally used to describe the self-advection velocity of two identical point vortices (of opposite sign) in an

otherwise still fluid (Kundu et al., 2015). This dual finding by Nøst and Børve (2021) naturally points to the fact that the jet strength gives the eddy strengths. In fact, assuming point vortices of equal strength, each having vorticity which scales as $U_{max}/a$, gives circulation $\Gamma = \pi U_{max}a$ (after applying Green's theorem) and a self-advection velocity $U_{dip} = (a/b)U_{max}$, i.e. the same as if assuming passive advection by the mean jet. This result indicates that the jet and the eddies are tightly connected and should probably not be studied in isolation. We will not dive into a further quantification of these relationships here but

only point back to Figures 8 and 9, which show how the jet itself is also 'stretched' by the dipoles, following their track away from the constriction.

### 4.3 Variable pressure gradient force at the exit

The observed dynamical behaviour on the downstream side of the constriction deviates significantly from what was reported by Hench et al. (2002) and Vennell (2006). In our results, we find a quasi-stationary pressure-gradient force directed toward the

constriction on the upstream side, as did the cited studies. But the pressure gradient is non-stationary on the downstream side due to the strong pressure fluctuations associated with the eddies. These oscillations vary over several minutes and may easily pass undetected by a point observation from the field if the data is smoothed over long intervals. They may also be 'undetected' in a numerical model if the eddy viscosity used is too high. Our interpretation is, however, in agreement with results reported by Nicolau del Roure et al. (2009), who studied the development of eddy dipoles in the laboratory. They concluded that dipoles

developed for some time close to the constriction before, as they put it, being entrained by the jet and subsequently advected downstream. Here we have also found indications that the same chain of events repeats to produce a set of secondary eddies and that the combined effect is a strong and chaotic flow field downstream of the constriction.



### 4.4 Limitations of this work

Shallow and narrow channels with strong tidally-driven currents are often well mixed. In addition, wind work and convective
mixing add to weak density stratification, especially during late winter months at high latitudes (Sælen, 1950). And indeed,
the ADCP in Tromsøysund revealed a barotropic tidal flow here, motivating the use of the 2D modelling approach. However,
we note that neither our observations nor model results are necessarily well-suited to assess the dynamical evolution when
freshwater input from rivers is more substantial in late spring, summer and fall. A strongly stable density stratification may
then allow the barotropic tide to excite internal gravity waves, which can then radiate energy far away from the constriction
(as discussed by e.g. Inall et al., 2004). A full 2D approach can even be questioned for a completely barotropic fluid as non-
hydrostatic vertical motions, neglected here, can be significant given sufficiently small scales and strong currents (Albagnac
et al., 2014; van Heijst, 2014). So it is conceivable that our study has ignored 3D effects that influence the eddy dynamics even
during wintertime.

Finally, it should be noted that we tuned the Smagorinsky mixing coefficient to obtain model results that fit well with the
observations. By increasing or decreasing this parameter, the model results changes somewhat. From the idealized results, it
may be tempting to conclude that the first dipole will always be stronger than the subsequent ones. However, in simulations
where we tested with stronger forcing and weaker horizontal viscocity (smaller Smagorinsky coefficient), we found that the
subsequent dipoles can approach the magnitude of the initial dipole. We can, therefore, not categorically state that the secondary
eddies are weaker than the first every single tidal cycle. But, more generally, applying higher eddy viscosity resulted in a flow
with weaker non-linearity and eddy production, producing velocity signals of lower frequency on the downstream side of
the constriction. This is, of course, not unexpected and only serves to highlight the point that the high-frequency variability
observed in Tromsøysund is inherently due to nonlinear dynamics.

## 5 Conclusions

A high-resolution low-viscosity numerical ocean model has been used to interpret high-frequency barotropic velocity fluc-
tuations observed by an ADCP downstream of a channel constriction in Northern Norway. The simulations showed that the
fluctuations, which exist only on the outflowing tidal phase, reflect flow-separation eddies that pinch off the channel walls and
propagate downstream. These eddies, which may form a propagating dipole pair, guide a jet between them. The variability
observed at the location of the ADCP reflects the velocity field of the eddies as they propagate downstream and lead to the
meandering of the jet.

Our analysis demonstrates that the pressure gradient force immediately downstream of the constriction is unstable, oscil-
lating between positive and negative values. This starkly contrasts with the quasi-steady situation expected from the classical
Bernoulli description of flow through constricted channels. As the flow-separation eddies spin up in cyclostrophic balance, they
generate pressure minima comparable to the larger-scale dynamic pressure field along the channel. These pressure anomalies
can thus remove the adverse pressure gradient downstream of the constriction, violating the original condition required for
flow separation. Once flow separation is no longer possible, the eddies detach and start to translate downstream. Finally, with




this initial dipole pair removed from the original generation zone, the adverse pressure gradient can again build up to generate secondary flow-separation eddies.

Clearly, both eddies and jets are strongly nonlinear. Both phenomena reflect flow separation and are, thus, in some sense, two sides of the same coin. The behaviour seen in this study is consistent with the conduct described in laboratory experiments by Nicolau del Roure et al. (2009), which indicates that the eddies grow to interact with the jet significantly before they shed off. After the eddies detach and form a dipole pair, the jet is typically guided between them. Indeed, the pressure gradient they generate is comparable in magnitude to the larger-scale pressure gradient force generated by constriction due to the Bernoulli effect. So the dipoles may be considered 'propagating constrictions' after they detach and move downstream. The current study does not pretend to offer a complete description of this complex flow dynamics. But focusing on the pressure signal might be a useful alternative to the more traditional focus on vorticity dynamics. The study has also highlighted the need for high-frequency field sampling, down to a few minutes, to study the life cycle of flow-separation eddies in tidal straits.

The dynamics studied here should be relevant beyond the academic community, not the least from the perspective of transporting chemical and biological material through the coastal zone. As outlined in the introductory section, the asymmetry between inflow and outflow essentially sets up an efficient Reynolds tracer flux through a constricted channel. The net transport capacity through any given channel is determined by how flow-separation eddies are generated and propagated there. Details of the tidal strength and coastal geometry are ultimately determining factors. The velocity field of flow-separation eddies and jets should also be considered by planners of tidal power plants and coastal communities that plan to modify the coastline. Typically, coastal land reclamation projects artificially introduce tighter constrictions. When assessing the environmental influence of such projects, one should not simply evaluate the flow speed increase expected due to the channel's changed Bernoulli effect but also the possible increase in high-frequency velocity fluctuations—which, as seen here, may manifest themselves far downstream of the constriction itself.

Finally, it is worth noting that the vorticity generated in tidal constrictions like the one studied here may interact with the Eulerian mean circulation. Børve et al. (2021), for example, show how vorticity fluxes across closed isobaths, e.g. around islands or banks, can drive a net residual circulation along these isobaths. Most fjords have a shallow sill at their entrance, with a deep basin inshore of the sill. It is, therefore, conceivable that the turbulent vorticity fluxes generated by tidal flow across such a sill can influence the time-mean residual circulation of the entire fjord system if the flow-separation eddies survive long enough to interact with closed bathymetric contours within the fjord.

**Competing interests**

The authors declare that they have no conflict of interest.

**Code availability**

Model code is available from https://github.com/UK-FVCOM-Usergroup/uk-fvcom (uk-fvcom branch).



**Data availability**

Model and observational data is available on request.

**Author contribution**

HE sat up the numerical model, conducted the numerical experiments and analyzed the data. All authors contributed to discussing and interpreting the results. HE and PEI wrote the initial draft, and all authors contributed to editing the paper.

**Acknowledgements**

This project is funded by the Research Council of Norway under contract 308796. The observations were obtained from Akvaplan-niva AS. We have used sea surface height fields from the TPXO 7.2 assimilated tidal model as boundary conditions

for our model simulation and gratefully thank the authors for releasing the model data publicly. Bathymetry data were provided by the Norwegian Mapping Authority.



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
