# Peer review of "Observations and modelling of tidally generated high-frequency velocity fluctuations downstream of a channel constriction"

_EGUsphere, 2023_

## Author Comment (AC2)

**Response to reviewer 2**

August 29, 2023

**In the model description, the authors mention the parameterization of the bottom friction coefficient Cd (line 75). However, it is not specified what D represents in this context. I presume it refers to Manning's n formulation. It would be beneficial to provide a reference or an explanation regarding the utilization of these coefficients.**

Thank you for pointing this out; you are correct in assuming it relates to Manning's n formulation. More specifically, the bottom friction coefficient $C_d$ is computed using the Chézy-Manning formulation;

$$C_d = g \frac{n^2}{\sqrt[3]{D}} \tag{1}$$

Where g is the earth's gravitational acceleration, $n$ is the Manning coefficient (which, by default in FVCOM, is set to 0.02), and D is the dynamic water column thickness. We will write the definition of $C_d$ in this form in the revised version of the manuscript.

This formulation is based on the assumption that the water column depth can approximate the hydraulic radius. It's unclear how true this is in the open ocean, but we understand it is commonly used in coastal ocean circulation models. We notice from other publications that $n \in [0.013, 0.023]$, with most reported choices of $n$ being in the higher end of the range (see, e.g. Blakely et al., 2022; Lee et al., 2020; Lyu and Zhu, 2018; Mayo et al., 2014; Kerr et al., 2013).

**If D represents depth, taking into account the U-shaped channel configuration, there would be more friction at the channel's edges compared to the center. How does this parameterization influence the formation of eddies? Does the flow structure remain unchanged, and do the article's conclusions hold true if the coefficient is assumed to be constant? I believe this aspect could be added to the discussion or the section on "Limitations of this work."**

It is important to note that the bottom roughness coefficient varies with depth (D) as $C_d \propto 1/\sqrt[3]{D}$. Bottom friction, as formulated in the shallow water equations, is also depth dependent; $F \propto C_d/D$. The depth dependence of friction is thus not dominated by the choice of drag coefficient. While the parameterization influences the formation of eddies, the fundamental physical mechanism still relies on stronger flow attenuation near the side walls, leading to flow separation and the dynamical evolution as described in our manuscript. We therefore expect that the dynamical system will exhibit a very similar behaviour if keeping the drag coefficient constant. We will incorporate this discussion into the revised version of the manuscript.

**Despite the intricate nature of fjords and straits, the authors have succeeded in relatively well modelling tidal dynamics in the study area. However, when comparing the tidal characteristics of the model with observations, the authors do not present a phase comparison. This comparison is essential for a comprehensive validation of the tidal model across the entire area. Since**

this study focuses on a single channel, the main objective of the setup is to establish accurate boundary conditions specific to this channel. Thus, it would be appropriate to restrict the comparison of model phases with observations to the channel area.

We agree with the reviewer's suggestion to compare the modelled sea surface elevation (SSE) to observations on either side of the channel. However, we do not have access to SSE measurements on the southern side of the channel constriction, just far into the nearby fjord (which also connects to other straits). If we had measurements near the choke point, we would have been able to assess whether the tidal forcing of the channel – the SSE difference across the choke point – was well captured.

However, Figure 7b shows that the modelled SSE variability over the period we model fits the observed variability, which together with the fact that the modelled- and observed speeds were similar indicates that the potential energy made available by the tide for driving the tidal jet should be reasonably well captured by the model. We therefore think a phase plot would be redundant, given the reasonable match with observations as shown in Fig 7 b.

**Regarding line 161, the statement "..by some earlier..." needs clarification. Which specific reference is being referred to?**

Thanks for pointing this out. In this sentence, we refer to Hench et al. (2002); Hench and Luettich (2003); Vennell (2006), and we will change the manuscript accordingly.

**References**

Blakely, C. P., et al., 2022: Dissipation and bathymetric sensitivities in an unstructured mesh global tidal model. *Journal of Geophysical Research: Oceans*, **127 (5)**, e2021JC018 178.

Hench, J. L., B. O. Blanton, and R. A. Luettich Jr, 2002: Lateral dynamic analysis and classification of barotropic tidal inlets. *Continental Shelf Research*, **22 (18-19)**, 2615–2631.

Hench, J. L. and R. A. Luettich, 2003: Transient tidal circulation and momentum balances at a shallow inlet. *Journal of Physical Oceanography*, **33 (4)**, 913–932.

Kerr, P., et al., 2013: Us ioos coastal and ocean modeling testbed: Evaluation of tide, wave, and hurricane surge response sensitivities to mesh resolution and friction in the gulf of mexico. *Journal of Geophysical Research: Oceans*, **118 (9)**, 4633–4661.

Lee, J., J. Lee, S.-L. Yun, and S.-K. Kim, 2020: Three-dimensional unstructured grid finite-volume model for coastal and estuarine circulation and its application. *Water*, **12 (10)**, 2752.

Lyu, H. and J. Zhu, 2018: Impact of the bottom drag coefficient on saltwater intrusion in the extremely shallow estuary. *Journal of Hydrology*, **557**, 838–850.

Mayo, T., T. Butler, C. Dawson, and I. Hoteit, 2014: Data assimilation within the advanced circulation (adcirc) modeling framework for the estimation of manning's friction coefficient. *Ocean Modelling*, **76**, 43–58.

Vennell, R., 2006: Adcp measurements of momentum balance and dynamic topography in a constricted tidal channel. *Journal of Physical Oceanography*, **36 (2)**, 177–188.

---

## Author Comment (AC3)

**Response to Reviewer 3**

August 29, 2023

In this response, we give short replies to the reviewer's comments and suggestions. The reviewer's points 1, 2, 5(i), 5(ii), 7, 9, 10, 11, 12, 13, 14, 15, 16, 20, 22, 23, 24, 25, 26, 27, 28(i) and 30 point to typos and suggested figure improvements. These will all be addressed in the revised version of the manuscript, but we will not address them specifically here.

**3. It is referenced to a small degree in the discussion, but I think it would be nice if the authors could add greater discussion surrounding i) the choice of running the model in 2D, and ii) how the results of a 3-D simulation might differ.**

Why did we choose to write the paper based on results from a 2D model?

- The velocity signal from the ADCP followed tidal frequencies closely, and with only a relatively weak depth dependence. These indicate that the flow is governed by sea surface elevation induced pressure gradients, i.e. barotropic dynamics.

- Having identified that the flow was quite barotropic, the cheapest (and simplest) next step is to use a 2D approach.

- After tuning the 2D model's horizontal diffusion for fit with the observations, the velocity signal matched the observations well.

- With the choice of parameters yielding best fit with the observations, the dynamical evolution in the channel did not resemble dynamical explanations we could find in the literature (see, e.g. Hench and Luettich (2003) – even though or fundamental assumptions were the same.

Hence; The observations motivated the use of a simple 2D approach. The results from the 2D model motivated a thorough analysis of the dynamical evolution, since the results differed significantly from previous similar studies. Presenting an alternative view of the fundamental dynamics using mathematically similar 2D barotopic dynamics felt like a logical step forward. Using a full 3D barotropic simulation would have introduced other uncertainties such as whether the model had sufficient vertical resolution, if the horizontal viscosity could be tuned to sufficiently low levels and retain numerical stability. We will add a truncated version of this to the discussion in the revised version of the manuscript.

A full 3D simulation is an obvious next step. It will open for assessments of processes we could not account for in our simplified setup may influence the evolution of eddy dipoles in tidal channels. We expect that a 3D simulation will differ from the simplified 2D view in several ways

- A boundary layer near the bottom represents a less direct energy loss from the bulk of the water column to bottom friction than implicitly assumed in a 2D model.

- A vertically sheared flow. We expect a mass flux toward the centre of the eddies near the bottom and away from the centre elsewhere – influencing the eddies longevity.

- Hydraulic jumps in stratified runs, consuming a significant portion of the energy provided to the flow by the tide rendering it unavailable for the eddies.

However, we speculate that such 3D effects, such as the barotropic effects reported by Albagnac et al. (2014), primarily influences the eddies' longevity on timescales greater than those we explore in this study (>1 hour). Furthermore, whether the physical interpretation presented herein will work for a 3D setting with the addition of realistic atmospheric forcing (surface winds and waves) would be speculative to comment on at the time of writing. We will further develop the "limitations of this study" section in the manuscript with this in mind.

However, our model results demonstrate that at this *specific* location, the 2D mode of the system dominates. We speculate that a full 3D model would not add processes that would significantly alter the flow for this specific case, and our conclusions regarding the temporal development of dynamical processes at the constriction.

**4. L87: "the model performed better". Please elaborate on what 'better' means/how it was quantified.**

"Better" in that sentence means that the tidal signal was in closer agreement with observations when forcing the model with TPXO7.2 rather than with the more recent version, TPXO9-atlas-v5. The modeled flow was too strong in Tromsøysund when using the TPXO9-atlas-v5 as forcing, and the sea surface elevation was out of phase with the tidal current. In contrast, the observations indicate that the flow and sea surface elevation are in phase (max flow speed at max tidal surface elevation). This likely shows that the tidal dynamics in the Tromsøysund are very sensitive to intricacies with the external forcing. The fjord connecting to Tromsøysund, Balsfjord, is connected to the open ocean at two distinct locations – in the south at 69.57 degrees north and in the north at 70.31 degrees north, exposing it to how well the model reproduces the Kelvin wave propagating northward, and spatial variations of the boundary conditions quality. We will modify the revised text with this in mind.

**6. Fig. 3, right panel: Other than the M2 tide, the constituents appear to follow more flat lines than a 1:1 slope. This is, for example, most apparent for the K1 tide. Why are you not able to capture the regional variability in the smaller-amplitude tides like you are with the M2 tide? How significant is this discrepancy for your tidal channel? (It is briefly mentioned starting on L95, but I would appreciate more analysis).**

This might seem a bit cliche coming from ocean modellers, but the rather flat curve for the minor tide components may indicate a problem with the observations, not the model. We believe these plots indicate that while the rigs were deployed sufficiently long to get a good estimate of the leading harmonic (M2), roughly one month was insufficient to get a good spectral estimate for the minor components. The minor components are much weaker than the M2 tide, and are more susceptible to noise. We will discuss this in further detail in the revised version of the manuscript.

**8. Fig 5: Is the 'm below sea level' relative to a mean SSH, or is it dynamic? (i.e., does the figure take tides into account). Would it make more sense to refer to height above sea floor to have a constant reference point?**

The z-axis is relative to depth below mean sea surface level. The instrument collects data from bins at constant depth relative to the sea floor, hence it makes sense to present the data relative to mean SSH. We will clarify that the data is presented relative to mean SSH in the revised manuscript.

**17. The jet in Fig. 9 appears to be angled toward the right. Why? Is it simply because of the Coriolis force, or are there other factors involved?**

This is an interesting point which was the subject of several discussions while we wrote this paper. We left it out of the manuscript to keep our focus on the initial development of the eddies, rather than the subsequent advection of the dipoles.

In our results, the cyclonic eddies generally "live" longer than anti-cyclonic eddies. The cyclonic eddies grow to sizes much greater than the constriction, and can introduce an eastward mean flow following the coastline, influencing the drift of the dipole. Whether this is the case in Fig. 9 is unclear.

One of our hypothesis on why cyclonic eddies are predominant, builds on the observation that *both* the cyclonic and anti-cyclonic eddies have low pressures in their centres. As they grow horizontally, the centripetal force decreases. In cyclonic eddies, the Coriolis force works in the same direction as the centripetal force, which combine to balance the pressure gradient force. In anti-cyclonic eddies, however, the Coriolis force and the pressure gradient force work against the centripetal force. This *may* explain why the anti-cyclonic eddies dissipate earlier than the cyclonic ones, but we haven't looked into this in more detail, or done a proper literature search to see if others have had the same idea.

**18. Figure 10, with its clear message and simplicity is beautiful!**

Thank you!

**19. Fig 11 and elsewhere: m/s2 are units of acceleration, not a force. Please clarify terminology.**

Good point. We will clarify that we express force per unit mass, i.e. acceleration.

**21. I think there is a word missing from L240: 'on southward and inward what?' Flow?**

Thank you for pointing out this typo; the sentence should say "on southward flow".

**28 (ii): L352 on: I appreciate this nice, practical finish to the manuscript.**

Thank you!

**29. There is discussion around the alternating rotation of the eddies as well as the role mixing plays within the channels. Are the eddies large enough to lead to local differences in vertical mixing on either side of the channel (because of their alternate rotation), or would that be small compared to background mixing?**

We are a little unsure of what the reviewer is referring to here. Presumably, the vortex motion will impact the exact lateral distribution of (shear-generated) turbulent mixing. It is an interesting idea, and we might flag this in the revised text (under discussion), but we admit that anything we could say on the topic would be speculative.

**References**

Albagnac, J., F. Y. Moulin, O. Eiff, L. Lacaze, and P. Brancher, 2014: A three-dimensional experimental investigation of the structure of the spanwise vortex generated by a shallow vortex dipole. *Environmental Fluid Mechanics*, **14 (5)**, 957–970.

Hench, J. L. and R. A. Luettich, 2003: Transient tidal circulation and momentum balances at a shallow inlet. *Journal of Physical Oceanography*, **33 (4)**, 913–932.

---

## Author Response (AR1)

**Final author reply to the editor**

September 22, 2023

————————————

Dear Editor.

The reviewer comments were all useful and well received. Below we list all reviewer comments, address them and refer to where we have made changes to the manuscript.

**1  Reviewer 1**

- My only suggestion refers to the figures. I think all the figures are helpful and required but there are some inconsistencies between figures in terms of style that could be addressed. Figures 3b, 6, 7, 11 and 13 could be drawn with the same style. Include variable name with the units on the y axis and remove title (as appropriate). Increase the font size of labels and axis ticks. Remove shaded background but keep some of the grid lines (enough to be able to compare between figures).

    - **We have updated all figures according to the suggestions, except for Fig 1. and Fig. 14.**

**2  Reviewer 2**

- In the model description, the authors mention the parameterization of the bottom friction coefficient Cd (line 75). However, it is not specified what D represents in this context. I presume it refers to Manning's n formulation. It would be beneficial to provide a reference or an explanation regarding the utilization of these coefficients.

    - **The reviewer is correct in assuming it relates to Manning's n formulation. More specifically, the bottom friction coefficient $C_d$ is computed using the Chézy-Manning formulation;**

    $$C_d = g \frac{n^2}{\sqrt[3]{D},} \tag{1}$$

    **where $g$ is Earth's gravitational acceleration, $n$ is the Manning coefficient (which, by default in FVCOM, is set to 0.02), and $D$ is the dynamic water column thickness. We have updated the definition of $C_d$ to this form in the revised version of the manuscript. The formulation is based on the assumption that the water column depth can approximate the hydraulic radius. It's unclear how true this is in the open ocean, but we understand that it is commonly used in coastal ocean circulation models. We notice from other publications that $n \in [0.013, 0.023]$, with most reported values being in the higher end of the range (see, e.g. Blakely et al., 2022; Lee et al., 2020; Lyu and Zhu, 2018; Mayo et al., 2014; Kerr et al., 2013). We have now also added this description to the manuscript, see L75 - L79.**

    .

- If D represents depth, taking into account the U-shaped channel configuration, there would be more friction at the channel's edges compared to the center. How does this parameterization influence the formation of eddies? Does the flow structure remain unchanged, and do the article's conclusions hold true if the coefficient is assumed to be constant? I believe this aspect could be added to the discussion or the section on "Limitations of this work."

  - **It is important to note that the bottom drag coefficient in this formulation is inversely proportional to the cube root of depth, i.e. $C_d \propto 1/\sqrt[3]{D}$. The effects of bottom friction in the shallow water equations is then inversely proportional to depth, i.e. $F \propto C_d/D$. The depth-dependence of friction is thus not primarily dominated by the relatively weak depth-dependence in the drag coefficient. While the parameterization influences the formation of eddies, the fundamental physical mechanism still relies on stronger flow attenuation near the side walls, leading to flow separation and the dynamical evolution as described in our manuscript. We therefore expect that the dynamical system will exhibit a very similar behaviour if keeping the drag coefficient constant. We have incorporated this discussion into the revised version of the manuscript, see L346 - L351**

- Despite the intricate nature of fjords and straits, the authors have succeeded in relatively well modeling tidal dynamics in the study area. However, when comparing the tidal characteristics of the model with observations, the authors do not present a phase comparison. This comparison is essential for a comprehensive validation of the tidal model across the entire area. Since this study focuses on a single channel, the main objective of the setup is to establish accurate boundary conditions specific to this channel. Thus, it would be appropriate to restrict the comparison of model phases with observations to the channel area.

  - **We agree with the reviewer's suggestion to compare the modelled sea surface elevation (SSE) to observations, ideally on either side of the channel. However, we do not have access to SSE measurements on the southern side of the channel constriction, other than far into the nearby fjord (which also connects to other straits). If we had measurements near the choke point, we would have been able to assess whether the tidal processes in the channel – leading to SSE differences across the choke point – were well captured.**

    **We, however, do not believe such a detailed comparison is essential. Figure 7b shows that the modelled SSE variability over the period we present in this paper fits the observed variability. Together with the fact that the modelled and observed speeds were similar, this indicates that the potential energy provided by the tide for driving the tidal jet should is reasonably well captured by the model. This, in turn, leads us to conclude that the model results are of sufficient quality to study the high-frequency oscillation flow phenomenon, at least at a process identification level—which is the aim of this work.**

- Regarding line 161, the statement "..by some earlier..." needs clarification. Which specific reference is being referred to?

  - **Here we now refer to Hench et al. (2002); Hench and Luettich (2003); Vennell (2006). The manuscript has been changed accordingly, see L178**

**3   Reviewer 3**

- ADCP is not defined until L57, but the acronym is used earlier (abstract and L18). Please define earlier.

- **We now define the acronym in the introduction, L18-L19. Consequently, we also removed the definition of the acronym on L57**

- L44: Should be 'exCited', not exited.

  - **Fixed, see L44**

- It is referenced to a small degree in the discussion, but I think it would be nice if the authors could add greater discussion surrounding i) the choice of running the model in 2D, and ii) how the results of a 3-D simulation might differ.

  - **Why did we choose to use a 2D approach?**
    * **The velocity signal from the ADCP followed tidal frequencies closely, and with only a relatively weak depth dependence. These indicate that the flow is governed by sea surface elevation induced pressure gradients, i.e. barotropic dynamics.**
    * **Having identified that the flow was quite barotropic, the cheapest (and simplest) next step is to use a 2D approach.**
    * **After tuning the 2D model's horizontal diffusion for fit with the observations, the velocity signal matched the observations well.**

    **With the choice of parameters yielding best fit with the observations, the dynamical evolution in the channel did not resemble dynamical explanations we could find in the literature (see, e.g. Hench and Luettich (2003) – even though or fundamental assumptions were the same.**

    **Hence; The observations motivated the use of a simple 2D approach. The results from the 2D model motivated a thorough analysis of the dynamical evolution, since the results differed significantly from previous similar studies. Presenting an alternative view of the fundamental dynamics using mathematically similar 2D barotopic dynamics felt like a logical step forward. We have developed the discussion with a truncated version of these arguments, see L327 - L333**

  - **How might the results of a 3-D simulation differ?**

    **A full 3D simulation is an obvious next step and would open for assessments of processes we could not account for with the shallow-water formulation. We expect that a 3D simulation will differ from the simplified 2D view in several ways:**

    * **A boundary layer near the bottom represents a less direct energy loss from the bulk of the water column to bottom friction than implicitly assumed in a 2D model.**
    * **A vertically-sheared flow. We expect a mass flux toward the centre of the eddies near the bottom and away from the centre elsewhere – possibly influencing the eddies longevity.**
    * **The possibility for hydraulic jumps in stratified runs, consuming a significant portion of the energy provided to the flow by the tide—rendering it unavailable for the eddies.**
    * **Influence of internal pressure gradients associated with e.g. internal tide generation.**

    **However, we speculate that such 3D effects, such as the barotropic effects reported by Albagnac et al. (2014), primarily influences the eddies' longevity on timescales greater than those we explore in this study (>1 hour). Furthermore, whether the physical interpretation presented herein will work for a 3D setting with the addition of realistic atmospheric forcing (surface winds and waves) would be speculative to comment on at the time of writing. We have modified the discussion with a truncated version of these arguments, see L333 - L345**

- L87: "the model performed better". Please elaborate on what 'better' means/how it was quantified.

  – **"Better" in that sentence means that the tidal signal was in closer agreement with observations when forcing the model with TPXO7.2 rather than with the more recent version, TPXO9-atlas-v5. We have updated the text on L92 - L93 with this clarification.**

- Please add missing/appropriate labels (axes, colorbars, and/or titles) to all figures as necessary. Providing only the units on the axis and variable name in the caption is difficult for the reader to take in the meaning of the plot.

  – **We have modified the font size, colorbars and variable name in the axis of all figures other than Fig. 1 and Fig. 14.**

- Fig. 3, left panel: Some of the symbols are a bit hard to make out on the map. Consider using colours that stand out more against the teal?

  – **We have added black edge colors to the symbols in the left panel of Fig. 3 to make them stand out more clearly from the teal.**

- Fig. 3, right panel: Other than the M2 tide, the constituents appear to follow more flat lines than a 1:1 slope. This is, for example, most apparent for the K1 tide. Why are you not able to capture the regional variability in the smaller-amplitude tides like you are with the M2 tide? How significant is this discrepancy for your tidal channel? (It is briefly mentioned starting on L95, but I would appreciate more analysis).

  – **This might seem a bit of a cliche coming from ocean modellers, but the rather flat curve for the minor tide components may indicate a problem with the observations, not the model. It is possible that the strong tidal current in this region have tilted the rig, introducing artificial variability to the pressure signal. It is also possible that while the rigs were not deployed for sufficiently long to get a good estimate of the leading harmonic (M2), roughly one month was insufficient to get a good spectral estimate for the minor components. Regardless of explanation model, the minor components are much weaker than the M2 tide, and are more susceptible to noise. We have added a discussion around this to the manuscript, see L96 - L116.**

- Fig 4: It might be nice to add a middle panel here showing what the channel looks like at an intermediate scale. As it stand it is difficult to get a sense for the entire channel from the full domain and zoomed-in domain. Consider also using mesh lines other than green. Because of the increased resolution near the channel, the domain looks shallower in the left panel.

  – **We have updated Fig. 4 with a middle-panel to show the channel at an intermediate scale**

- Fig 5: Is the 'm below sea level' relative to a mean SSH, or is it dynamic? (i.e., does the figure take tides into account). Would it make more sense to refer to height above sea floor to have a constant reference point?

  – **The z-axis is relative to depth below mean sea surface level. The instrument collects data from bins at constant depth relative to the sea floor, hence it makes sense to present the data relative to mean SSH. We have updated the y-label in Fig. 5 accordingly.**

- Fig. 7, 11, 13. Because of the grey background on the axes, the grey shading is not obvious. Please consider removing the grey background, or choosing a different colour for shading.

– **We have removed the grey background in Fig. 3, 4, 6, 7, 11, 13**

- Fig 7 caption: 'studies' -¿ 'studied'

  – **We have changed the caption accordingly, see the caption of Fig. 7.**

- L136: remove the hyphen between nine-time

  – **Removed, see L153**

- Fig. 8: the white star is difficult to see in some of the panels (e.g., A-C) due to the pale yellow. Consider outlining the star, using a different colour, or only plotting it in one panel.

  – **We have re-drawn Fig. 8 so that the edge color of the white star is black, this helps it stand out more from the teal coloring.**

- Fig. 8: The lower panel with the time series could benefit by being made larger. It is difficult to see the different points and lines clearly otherwise.

  – **Good point. We chose to make the time-series shorter to make the points and lines easier to see.**

- L138: remove 'above' when referring to the snapshots (it is ambiguous if you mean above the text or above the lower panel, but is unnecessary)

  – **Removed.**

- L140: remove '(+15 mins)' as this is already clear.

  – **Removed.**

- L153: Consider replacing 'So a main take home...' with 'A key...' (or similarly concise phrase)

  – **We have changed the manuscript accordingly, see L170**

- The jet in Fig. 9 appears to be angled toward the right. Why? Is it simply because of the Coriolis force, or are there other factors involved?

  – **This is an interesting point which was the subject of several discussions while we wrote this paper. We left it out of the manuscript to keep our focus on the initial development of the eddies, rather than the subsequent advection of the dipoles.**

  – **In our results, the cyclonic eddies generally "live" longer than anti-cyclonic eddies. The cyclonic eddies grow to sizes much greater than the constriction, and can introduce an eastward mean flow following the coastline, influencing the drift of the dipole. Whether this is the case in Fig. 9 is unclear.**

  – **One of our hypotheses on why cyclonic eddies are predominant, builds on the observation that *both* the cyclonic and anti-cyclonic eddies have low pressures in their centres. As they grow horizontally, the centripetal force decreases. In cyclonic eddies, the Coriolis force works in the same direction as the centripetal force, which combine to balance the pressure gradient force. In anti-cyclonic eddies, however, the Coriolis force and the pressure gradient force work against the centripetal force. This *may* explain why the anti-cyclonic eddies dissipate earlier than the cyclonic ones, but we haven't looked into this in more detail, or done a proper literature search to see if others have had the same idea.**

  – **This phenomenon is interesting, but it occurs after the initial buildup of the eddies that we are studying in this manuscript. We therefore chose to leave this discussion out of the manuscript.**

- Figure 10, with its clear message and simplicity is beautiful!

  – **Thank you! We do appreciate this kind comment.**

- Fig 11 and elsewhere: m/s2 are units of acceleration, not a force. Please clarify terminology.

  – $m/s^2$ **are units of force pr. unit mass, often referred to as the specific force. We have clarified this in the revised manuscript, see the captions of Fig 10, 11, 12 and 13.**

- Fig 11 legend: along-SHORE pressure gradient

  – **No, that would not be correct. Here we have rotated the force-vector** *along the channel*, **as illustrated in Figure 2**

- I think there is a word missing from L240: 'on southward and inward what?' Flow?

  – **Thanks. This was a typo; the sentence should say "on southward flow", and we have changed the text accordingly (L258). We also changed L259 to be "northward flow"**

- L247: '. . . fluctuates between positive and negative values in quite a noisier manner. . . ' -¿ ". . . fluctuates noisily between positive and negative values. . . '

  – **We have modified the text accordingly, see L265**

- L248: 'So, despite the more noisy situation here. . . ' -¿ 'So, despite the increased noise. . . ' -¿

  – **We have modified the text accordingly (L266)**

- Fig 13 (caption): perhaps referring back to the idealized equivalent in F11 would help the reader interpret the plot?

  – **We have now referred to Fig. 11 in the caption of Fig. 13.**

- L253: remove hyphen in 'in-viscid'

  – **Fixed, L271**

- L320: 'changes' -¿ 'changed'

  – **Fixed. (L354)**

- L348: fix backwards apostrophe on quote of 'propagating constrictions'

  – **Fixed. (L79)**

- L348/349: Remove 'pretend to' from 'The current study does not pretend to offer a complete description of this complex flow dynamics.' While I appreciate you acknowledging the short comings, somehow that phrase seems to lessen the great work you have done!

  – **Thank you, we have now removed it in the revised manuscript. (L382)**

- L352 on: I appreciate this nice, practical finish to the manuscript.

  – **We appreciate the reviewer's kind comments.**

- There is discussion around the alternating rotation of the eddies as well as the role mixing plays within the channels. Are the eddies large enough to lead to local differences in vertical mixing on either side of the channel (because of their alternate rotation), or would that be small compared to background mixing?

- We are actually somewhat unsure of what the reviewer is referring to here. Presumably, the vortex motion will impact the exact lateral distribution of (shear-generated) turbulent mixing. It is an interesting idea, but we admit that anything we could say on the topic would be speculative. Our manuscript focus on 2D dynamics, and our data is not suitable to comment on dynamical processes acting in the vertical direction. We consequently have not modified the manuscript.

- L375: 'sat' -¿ 'set'

  - Fixed. (L408)

**References**

Albagnac, J., F. Y. Moulin, O. Eiff, L. Lacaze, and P. Brancher, 2014: A three-dimensional experimental investigation of the structure of the spanwise vortex generated by a shallow vortex dipole. *Environmental Fluid Mechanics*, **14 (5)**, 957–970.

Blakely, C. P., et al., 2022: Dissipation and bathymetric sensitivities in an unstructured mesh global tidal model. *Journal of Geophysical Research: Oceans*, **127 (5)**, e2021JC018 178.

Hench, J. L., B. O. Blanton, and R. A. Luettich Jr, 2002: Lateral dynamic analysis and classification of barotropic tidal inlets. *Continental Shelf Research*, **22 (18-19)**, 2615–2631.

Hench, J. L. and R. A. Luettich, 2003: Transient tidal circulation and momentum balances at a shallow inlet. *Journal of Physical Oceanography*, **33 (4)**, 913–932.

Kerr, P., et al., 2013: Us ioos coastal and ocean modeling testbed: Evaluation of tide, wave, and hurricane surge response sensitivities to mesh resolution and friction in the gulf of mexico. *Journal of Geophysical Research: Oceans*, **118 (9)**, 4633–4661.

Lee, J., J. Lee, S.-L. Yun, and S.-K. Kim, 2020: Three-dimensional unstructured grid finite-volume model for coastal and estuarine circulation and its application. *Water*, **12 (10)**, 2752.

Lyu, H. and J. Zhu, 2018: Impact of the bottom drag coefficient on saltwater intrusion in the extremely shallow estuary. *Journal of Hydrology*, **557**, 838–850.

Mayo, T., T. Butler, C. Dawson, and I. Hoteit, 2014: Data assimilation within the advanced circulation (adcirc) modeling framework for the estimation of manning's friction coefficient. *Ocean Modelling*, **76**, 43–58.

Vennell, R., 2006: Adcp measurements of momentum balance and dynamic topography in a constricted tidal channel. *Journal of Physical Oceanography*, **36 (2)**, 177–188.